# Realization of all-optical vortex switching in exciton-polariton condensates

Xuekai Ma [1✉], Bernd Berger[2], Marc Aßmann [2], Rodislav Driben[1], Torsten Meier [1], Christian Schneider[3], Sven Höfling [3,4] & Stefan Schumacher[1,5]

Vortices are topological objects representing the circular motion of a fluid. With their additional degree of freedom, the vorticity, they have been widely investigated in many physical systems and different materials for fundamental interest and for applications in data storage and information processing. Vortices have also been observed in non-equilibrium exciton-polariton condensates in planar semiconductor microcavities. There they appear spontaneously or can be created and pinned in space using ring-shaped optical excitation profiles. However, using the vortex state for information processing not only requires creation of a vortex but also efficient control over the vortex after its creation. Here we demonstrate a simple approach to control and switch a localized polariton vortex between opposite states. In our scheme, both the optical control of vorticity and its detection through the orbital angular momentum of the emitted light are implemented in a robust and practical manner.

[1] Department of Physics and Center for Optoelectronics and Photonics Paderborn (CeOPP), Universität Paderborn, Warburger Strasse 100, 33098 Paderborn, Germany. [2] Experimentelle Physik 2, Technische Universität Dortmund, 44227 Dortmund, Germany. [3] Technische Physik, Physikalisches Institut and Würzburg-Dresden Cluster of Excellence ct.qmat, Universität Würzburg, Am Hubland, 97074 Würzburg, Germany. [4] SUPA, School of Physics and Astronomy, University of St. Andrews, St. Andrews KY16 9SS, UK. [5] College of Optical Sciences, University of Arizona, Tucson, AZ 85721, USA
✉email: xuekai.ma@gmail.com

The two fundamental states of a vortex represent different rotation directions and can be used to encode binary information for example in vortex-based random access memories[1]. Switching a vortex between its two states is coveted by scientists and the dynamics have been investigated intensively[2–8]. For example, the manipulation of core magnetization of a vortex in ferromagnetic materials can be achieved by magnetic fields[2–4] or by an electrical current through the spin-transfer effect[9,10]. To overcome the so-called electric bottleneck in data transmission, however, all-optical networks have been proposed which requires optical data storage and processing including all-optical switching. Promising platforms to create and utilize vortices in optics[11–14] include plasmonic or dielectric metasurfaces[15–20] as well as exciton-polaritons in semiconductor microcavities[21–24].

A typical planar semiconductor microcavity is sketched in Fig. 1a. As fundamental exictations of this system, exciton-polaritons are hybrid light-matter objects, composed of both quantum well excitons and cavity photons. In a high-quality semiconductor microcavity, the strong coupling between photons and excitons leads to an anti-crossing of the fundamental modes generating well separated upper and lower polariton branches as illustrated in Fig. 1b. Through their photonic component polaritons can be optically excited and probed and show remarkable coherence properties. The excitonic component leads to strong optical nonlinearity, which is at the heart of many interesting nonlinear phenomena[25–29], including the formation of non-equilibrium polariton condensates[30,31] and polariton vortices[32–35]. This nonlinearity also allows for active optical control of the system dynamics, for example with tailored and spatially structured off-resonant excitation.

In the case of polariton condensation, an off-resonant optical beam creates a reservoir of hot excitons. Through Coulomb and phonon scattering events these then relax towards the so-called bottleneck region close to the top of the LPB. From there, polaritons undergo stimulated scattering towards the ground state as sketched in Fig. 1b. Due to the finite lifetime of polaritons, even for a macroscopic population and a buildup of long-range coherence in the polariton ground state (that is, condensation), the coherent polariton system intrinsically is in a non-equilibrium state subject to drive and decay. A persistent external pump with frequency far above the exciton resonance can be used to sustain the population in the condensate. Besides replenishing the condensate, however, the reservoir excitations also play an important role in the dynamics of the condensate due to reservoir–condensate interaction. Moreover, the reservoir inherits the spatial shape from the optical pump as indicated in Fig. 1a, which can be tailored to specific needs. Here, we use a ring-shaped pump that yields a reservoir-induced external potential seen by the condensate as illustrated in Fig. 1c (including the perturbation by an additional control beam). This potential is used to trap the condensate. As a result of spontaneous symmetry breaking in the persistently pumped nonlinear dynamical system, this ring-shaped excitation profile also leads to the formation of a vortex inside the ring. Independent from the off-resonant pump beam, this vortex carries a finite orbital angular momentum (OAM). For a system and optical excitation setup with rotational symmetry, two possible vortex states with opposite topological charges are supported by the same pump excitation. Recently, chiral polaritonic lenses[32] and external potentials have been proposed to generate specific vortices by breaking the central symmetry[34,36,37] or by optically breaking time-reversal symmetry[38,39], and indirect vortex control using complex optical setups was proposed theoretically[35,40].

Here, we demonstrate experimentally and theoretically the direct optical control of an optically imprinted trapped vortex charge using an ultra-short (120 fs in our experiments) off-resonant optical control pulse that can reverse the topological charge of an already formed vortex. The basic principle is that an off-resonant optical pulse close to the vortex excites a potential barrier, which stops the rotating vortex and flips it to the opposite rotation direction. Our detailed analysis shows that with the perturbation of the initially formed vortex mode the nonlinear dynamical system evolves into an oscillating dipole mode. After switching off the perturbation the system returns to the vortex mode with a charge that depends on the duration and the power of the control pulse. We directly measure the topological charge of the vortex and its temporal evolution using an efficient OAM sorting approach. Compared with interferometric approaches for which a phase reference is needed, this simple and robust detection scheme is directly relevant to future applications as it only requires the bare sample emission as the input and works down to the level of individual photons[41]. In the present work the vortex switching is demonstrated in a high-quality GaAs-based semiconductor sample, however, we find that the scheme is quite robust against sample disorder which will be important for the transfer to other materials, potentially allowing operation at up to room temperature.[42,43]

## Results

**Principle of optical vortex control.** Before moving on to the experimental realization, we will first explain the basic principle of vortex switching supported by a detailed numerical analysis. Our calculations are based on a microscopic model for the coupled spatio-temporal light-field exciton dynamics inside the

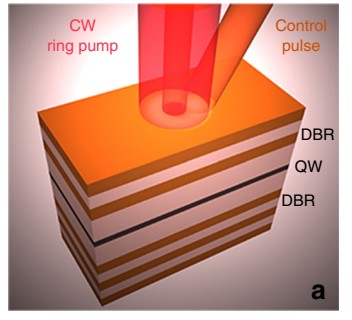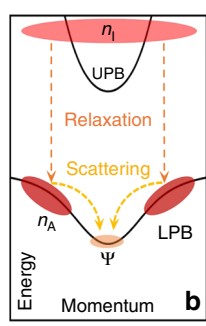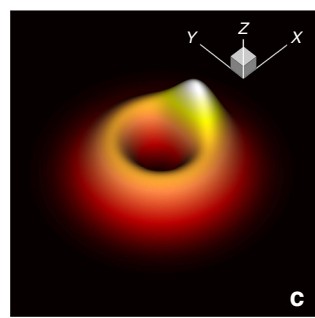

**Fig. 1 Optical setup and excitation of a polariton condensate. a** Sketch of a planar quantum-well (QW) based semiconductor microcavity with distributed-Bragg reflectors (DBR) at top and bottom. **b** Sketch of the energy versus in-plane momentum dispersions of the lower (LPB) and upper (UPB) polariton branches. As indicated in **a**, the inactive reservoir ($n_I$) is excited by an off-resonant continuous wave (CW) ring-shaped pump beam and an off-resonant Gaussian-shaped control pulse. Subsequently, relaxation into the active reservoir ($n_A$) and from there stimulated scattering into the condensate ($\Psi$) at the bottom of the LPB occur. **c** Three-dimensional sketch of the total optically induced external potential experienced by the coherent polariton condensate when the system is optically pumped as sketched in **a**.

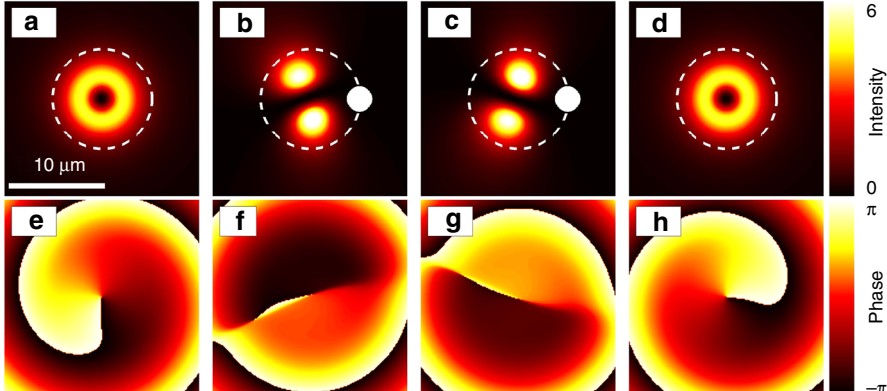

**Fig. 2 Principle of optical vortex control. a–d** Polariton density in μm$^{-2}$ and **e–h** phase distribution in radians of vortices for different excitation settings. The vortex in **a**, **e** forms for excitation with only the ring-shaped CW pump with a profile as indicated by the white dashed circle. When in addition to the ring-shaped CW pump a Gaussian control beam is switched on as indicated by the white spots, an oscillating dipole mode is created as shown in **b**, **f** and **c**, **g** at different points in time. **d**, **h** When the additional control beam is switched off again, the system returns to a vortex mode, however, with a rotation direction of the vortex that depends on the point in time at which the control is switched off. Here, panel **d** shows a successful vortex switching event with the vortex charge inverted after perturbation by the control beam as is apparent from the phase profiles in **e** and **h**. The gradient of the phase distribution defines the local flow direction of the condensate.

semiconductor microcavity. More details are given in the Methods section. Let us first consider excitation of the microcavity system with an off-resonant continuous wave (CW) ring-shaped pump beam. In the case of full rotational symmetry, the polariton condensate that forms in the annular potential will spontaneously form a vortex with clockwise or counter-clockwise rotation depending on initial fluctuations in the coherent polariton field acting as a seed in the condensation process[35]. We will address the influence of sample disorder and reduced rotational symmetry in detail in a separate section below. In Fig. 2a, e, the example of a spontaneously formed vortex rotating counterclockwise is shown. Now, in addition to the CW ring pump we also switch on a Gaussian pump spatially localized in the ring region. This perturbation intentionally breaks the rotational symmetry of the optically imprinted external potential seen by the condensate as sketched in Fig. 1c and the condensate assumes a dipole mode as shown in Fig. 2b, c, f, g. This dipole mode, however, is not static, but persistently oscillates back and forth between the two orientations in Fig. 2b, c, f, g as long as the Gaussian perturbation is present. Once the Gaussian control beam is switched off and the height of the potential barrier is reduced, the perturbation potential acting on the dipole becomes weaker and eventually the dipole transforms back into a vortex mode which may lead to a vortex rotating in the direction opposite from the initial vortex as shown in Fig. 2d, h. The final rotation direction depends on the rotation direction of the dipole at the time when the perturbation potential is sufficiently reduced in strength to allow for a stable vortex mode to form again and thus can be set by altering the duration or magnitude of the perturbation potential. We note that an oscillation of polaritons between opposite OAM modes was observed with pulsed excitation of an eigenmode superposition in a elliptical microcavity[44].

**Experimental realization of vortex switching**. Here, we closely follow the concept discussed in the previous section. The main experimental results are shown in Fig. 3. We first create a vortex by only applying a ring-shaped CW pump beam as sketched in Fig. 1a. In order to optimize the visibility of the vortex switching process, we employ a pump beam with a small intentional asymmetry, such that a slightly preferred direction of rotation for the vortex exists. To identify the vortex switching process we use our direct access to the OAM of the light emitted from the

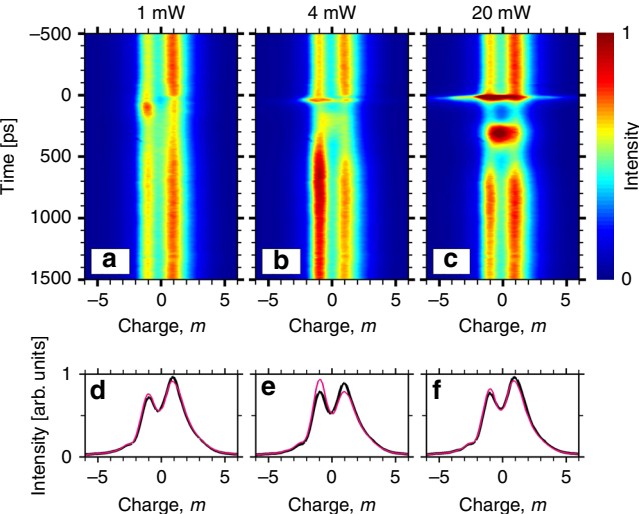

**Fig. 3 Experimental realization of optical vortex switching. a–c** Temporal traces of photoluminescence emitted from the polariton condensate, resolved into orbital angular momentum (OAM) components denoted as charge $m$. The same normalization is used for all data points. At small times for **a–c** only the ring-shaped off-resonant CW beam is on. In this case the signal is dominated by the $m = +1$ OAM component, emitted from a $+1$ vortex mode inside the condensate. At $t = 0$ ps a short 120 fs off-resonant control pulse with increasing power from **a** to **c** perturbs the free evolution of the vortex. After perturbation with a 1 mW control pulse in **a**, the condensate returns to the original $+1$ vortex state, that is, no vortex switching occurs. In **b**, for slightly increased control power of 4 mW, the vorticity in the condensate is switched to the $-1$ mode after the pulse has passed. In **c**, for further elevated control power of 20 mW, the vortex switching does not occur and the condensate returns to the $+1$ mode as in **a**. **d–f** 1D profiles corresponding to **a–c**, respectively, selected at different times. The thick black lines are at $t = -500$ ps (before switching pulse application) and the thin pink lines are at $t = 1500$ ps (after switching pulse application).

condensate on a shot-by-shot basis with picosecond time resolution. We cannot apply more common interferometric techniques in this case as phase patterns are not stable over several pulses and there is no phase reference for off-resonant excitation. Here, we employ OAM sorting[45,46], which converts the phase

gradient into a spatial displacement and is applicable down to the single-photon level. Figure 3a shows the measured time-resolved photoluminescence from the polariton condensate resolved into its OAM components (denoted as topological charge, $m$). Optical setup and data analysis[45,46] as well as sample details[47] are discussed in the Methods section. With excitation by a CW ring pump only, at time $t < 0$ ps the emitted signal is dominated by the $m = +1$ topological charge, evidencing that a counter-clockwise rotating vortex is created. At $t = 0$ ps an additional off-resonant pulse with duration of 120 fs is applied, creating spatially localized reservoir excitations such that the rotational symmetry of the effective external potential seen by the condensate is temporarily destroyed as discussed in the previous section. At low pulse power of 1 mW as shown in Fig. 3a, d, shortly after application of the pulse, at $t \sim 100$ ps the $m = +1$ state is suppressed and the $m = -1$ state dominates for a short period of time. Then the system reverts back to the $m = +1$ state at $t \sim 200$ ps. For a slightly increased pulse power of 4 mW as shown in Fig. 3b, the system oscillates back and forth twice with the $m = -1$ state dominating at $t \sim 50$ ps shortly after application of the pulsed perturbation and then persistently dominating the signal from $t \sim 300$ ps as the system switches into a new state with the formation of a clockwise rotating $m = -1$ vortex. In this case the application of the additional pulse inverts the vortex charge, resulting in switching between different vortex states. We note that the present experiments are not optimized to achieve maximum contrast but focus on the realization of the fundamental principle. A short discussion of the contrast in the OAM detection is given in the Supplementary Note 3. When the pulse power is increased further to 20 mW in Fig. 3c, f the system again undergoes an intermediate oscillation period with dominant $m = +1$ and $m = -1$ states but then at longer times returns to a state with predominantly counter-clockwise rotation of the vortex. Finally, we would like to note that while the duration of the control pulse in the experiments reported in Fig. 3 is only 120 fs, the effective duration of the perturbation is determined by rather complex relaxation processes. Accordingly, the switching dynamics reported here evolves on a much longer, few hundreds of picoseconds, time-scale. After having demonstrated vortex switching and control by simple application of a short off-resonant spatially localized light pulse in the present section, in the following section we will theoretically investigate the general dependence of the vortex dynamics on the power and duration of the control pulse.

**Vortex dynamics.** We write the intensity profile of the laser beam used for excitation of the microcavity system in the calculations as

$$P(r) = P_0 \frac{r^2}{w^2} e^{-\left(\frac{r^2}{w^2}\right)} + a P_0 e^{-\frac{(r')^2}{w_P^2}} e^{-\frac{t^2}{w_t^2}}, \quad (1)$$

with $r = \sqrt{x^2 + y^2}$ and $r' = \sqrt{(x - r_0)^2 + y^2}$. Here, we chose $P_0 = 6$ ps$^{-1}$ μm$^{-2}$. $w = 5$ μm is the radius of the ring-shaped CW pump beam, $a$ scales the intensity of the Gaussian pulse with spatial width $w_p = 3$ μm and peak intensity at $r_0 = 5$ μm, $w_t$ is the duration of the pulse.

In the calculations, we first create a vortex in the polariton condensate from initial noise by application of the off-resonant ring-shaped CW pump beam. After the condensate reaches stationary behavior in the vortex mode as in Fig. 2a, the Gaussian perturbation pulse with magnitude $a = 2$ is also applied. Figure 4a shows examples for the time evolution of the total topological charge $m$ of the condensate when the pulse is applied at $t = 0$ ps. The topological charge $m$ is calculated as the normalized angular

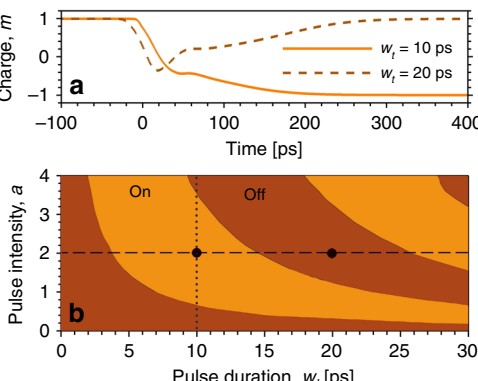

**Fig. 4 Control pulse dependence and system dynamics. a** Time evolution of the total topological charge of the polariton condensate for different duration of the control pulse with $w_t = 10$ ps and $w_t = 20$ ps, respectively, for fixed intensity with $a = 2$. For $w_t = 10$ ps successful vortex switching occurs whereas for $w_t = 20$ ps the charge returns to its original $+1$ state. **b** Computed final switching state depending on pulse duration $w_t$ and pulse intensity $a$. Orange regions represent successful switching, leaving the system in the on state. For brown regions the system returns to the off state after the control pulse has passed. Parameter sets used in panel **a** are indicated as black dots in panel **b**.

momentum expectation value

$$m(t) = \frac{L_z(t)}{N(t)}. \quad (2)$$

Here, $L_z(t) = -i \int \Psi^*(\mathbf{r}, t)(x\partial_y - y\partial_x)\Psi(\mathbf{r}, t) \, d\mathbf{r}$ is the average angular momentum and $N(t) = \int |\Psi(\mathbf{r}, t)|^2 \, d\mathbf{r}$ is the number of polaritons in the condensate. Results are shown for two different pulse lengths with $w_t = 10$ ps and $w_t = 20$ ps, respectively. For $w_t = 10$ ps, after application of the perturbation pulse the vortex with topological charge $m = +1$ is smoothly switched to the vortex state with opposite topological charge $m = -1$. For a longer pulse with $w_t = 20$ ps, however, the vortex is hindered in its rotation and temporarily converted into an oscillating dipole mode with near zero total topological charge and then it is stopped again and switched back to the original state with $m = +1$.

In a more complete picture, whether successful vortex switching (inversion of the vortex charge) occurs after application of the control pulse depends on both pulse duration and intensity as shown in Fig. 4b. For a fixed intensity of the Gaussian control pulse, the duration of the pulse directly affects the lifetime of the potential barrier induced. The lifetime of the potential barrier in turn determines how many times the intermediate dipole mode oscillates back and forth. If it is flipped an odd (even) number of times the final switching result is on (off). For a very long control pulse or CW control, the system persistently oscillates between the two topological states, topologically forming an oscillating dipole mode as shown in Fig. 2. For fixed duration, the switching dynamics and overall outcome depends on the intensity of the control. When the intensity is increased, a shorter pulse duration is required for vortex charge inversion. We find that for a higher potential barrier the oscillation period of the dipole is decreased. Moving along the vertical dotted line in Fig. 4b one can see that for fixed pulse duration the switching status alternates with increasing pulse intensity. This observation coincides well with the experimental results shown in Fig. 3. We note that in the experiment reported in Fig. 3, switching is achieved with a pulse duration of only 120 fs whereas in the simulations for such short pulses no switching would occur. In our theoretical description

the relaxation processes from the reservoir to lower energy states are only treated on a phenomenological level, in which the complicated electron–phonon interaction processes are not considered in detail. As a consequence, especially for short pulses, control pulse durations needed to achieve switching can not directly be compared between theory and experiment. The timescales of the switching dynamics (hundreds of picoseconds), however, in both experiments in Fig. 3 and theory in Fig. 4a agree well with each other.

**Role of disorder**. In this section we discuss the role that sample disorder plays for our results and analysis. In this context it is worth noting that in the measurements performed in the present work, for a fully rotationally invariant system, the vortex switching would not be evidenced. Data shown in Fig. 3 are obtained by averaging over many control pulses and switching events. If the rotation direction of the initial vortex state was only decided by random fluctuations triggering the condensation process, clockwise and counter-clockwise rotation would occur with equal probabilities. In that case the OAM-sorted signal of a single switching event would initially show only either one or the other of the $m = +1$ and $m = -1$ states. Due to averaging over many switching events, the measured signal would be distributed with equal intensity to the $m = +1$ and $m = -1$ states before and after incidence of the control pulse. So even if a switching process occurred, it would not be directly evidenced in the measured data. In the presence of a moderate disorder profile, however, rotational symmetry is intrinsically slightly broken, resulting in an initial imbalance of the $m = +1$ and $m = -1$ states and vortex switching can be clearly observed as shown in Fig. 3. Another benefit of the disorder leading to broken symmetry is that the system as shown in Fig. 3b will predominantly occupy the $m = +1$ state again on a long timescale (several nanoseconds after the application of the pulse). Otherwise, again a signal with intensity equally distributed to both $m = \pm 1$ states would be observed in Fig. 3b, as the vortex would get switched back and forth with every additional pulse. The built-in asymmetry enables us to observe the switching dynamics even when averaging over many events.

In Fig. 5a, we show the time-integrated spatially resolved photoluminescence before application of the control pulse at $t < 0$ in Fig. 3a. Figure 5b shows the corresponding ring-shaped intensity profile of the CW pump beam used to excite the sample that due to its imperfections also slightly breaks the system symmetry. Apparently, in the real system with its imperfections the condensate state formed in the system as shown in Fig. 5a does not show the perfect ring-shaped emission profile expected for an ideal individual vortex as assumed in the theoretical calculations, for example, shown in Fig. 2a. Not surprisingly, for each fixed point in time in Fig. 3 also the OAM-resolved photoluminescence does not show only a single OAM contribution of well defined topological charge but two dominant OAM states with different amplitudes of $m = +1$ and $m = -1$, respectively. With broken rotational symmetry, a condensate state with coexisting topological charges $m = +1$ and $m = -1$ is created. In the Supplementary Note 1 we include correlation measurements supporting this observation. The simultaneous presence of $m = +1$ and $m = -1$ components also explains the dipole-like appearance of the spatial intensity profile in Fig. 5a.

To mimic the experimental conditions more closely, in our numerical simulations, we now include a finite sample disorder as a random external potential for the coherent polariton condensate with realistic magnitude of 0.2 meV and correlation length 1 μm[47]. For one random disorder configuration, the resulting condensate density and phase are shown in Fig. 5c, d before and in Fig. 5e, f after application of a control pulse to switch the rotation direction of the vortex. We note that in the numerical simulations we find that if sample disorder gets too strong, a stable dipole mode is formed that can not serve as the initial state for switching. Experimentally, we find that a different or stronger local disorder landscape can lead to switching dynamics that is less stable. To illustrate our observations, vortex switching dynamics measured at different sample positions are included in Supplementary Note 2. Overall, this discussion illustrates that in the experimental observations in Figs. 3 and 5a, a vortex state with two OAM contributions but one dominant contribution is created and switched in the presence of sample disorder. Finally, we would like to note that the vortex switching dynamics could potentially also be observed in the emission spectrum of the condensate before and after application of the switching pulse. Experimentally, it was demonstrated that a broken rotational symmetry results in slightly different energies of vortex states with different rotation directions[38,48].

## Discussion

In the present work we have demonstrated the all-optical switching of the charge of a single localized vortex in a micro-cavity polariton condensate. The generation of the vortex and the switching is achieved using only off-resonant control pulses. In contrast to resonant excitation, our switching scheme is universal. With the same optical beam we can switch from an $m = -1$ state into an $m = +1$ state and vice versa, while for resonant excitation this is not as simple. In principle the same scheme also works for higher topological charges. Numerical results showing switching between topological charges $m = +2$ and $m = -2$ are included in the Supplementary Note 4. Using only off-resonant excitation, our approach may also allow for an electrically pumped realization in the future and our experimental readout of the vortex charge by OAM sorting is exceptionally simple and practical in contrast to an interferometric approach and works down to the single-photon level. We further show that our scheme is remarkably robust against sample disorder and system imperfections and may pave the way for practical realizations of all-optical switching and information processing based on polariton vortices in semiconductor microcavities.

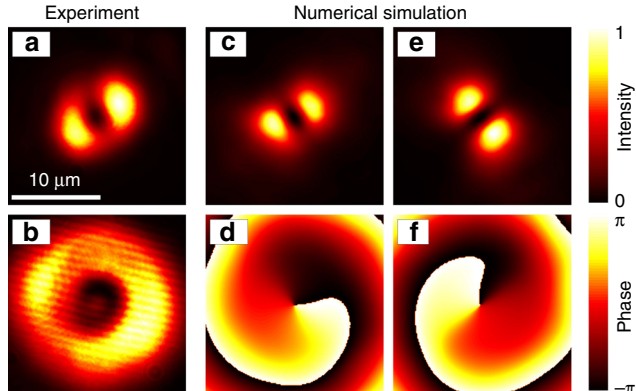

Experiment Numerical simulation

**Fig. 5 Vortices in experiment and numerical simulation. a** Real-space representation of normalized measured condensate photoluminescence as observed in Fig. 3 before application of the control pulse at negative times. **b** Normalized ring-shaped optical excitation intensity profile used in experiment. **c–f** Computed normalized condensate density and corresponding phase profiles in vortex states before (**c**, **d**) and after (**e**, **f**) switching including sample disorder.

## Methods

**Theory.** The dynamics of the coherent polariton system can be analyzed in detail based on a driven-dissipative Gross–Pitaevskii (GP) type model[49]

$$i\hbar\frac{\partial\Psi(\mathbf{r},t)}{\partial t} = \left[-\frac{\hbar^2}{2m_{\mathrm{eff}}}\nabla_\perp^2 - i\hbar\frac{\gamma_{\mathrm{c}}}{2} + g_{\mathrm{c}}|\Psi(\mathbf{r},t)|^2 \right.$$
$$\left. + \left(g_{\mathrm{r}} + i\hbar\frac{R}{2}\right)n_{\mathrm{A}}(\mathbf{r},t) + g_{\mathrm{r}}n_{\mathrm{I}}(\mathbf{r},t) + V_{\mathrm{ext}}(\mathbf{r})\right]\Psi(\mathbf{r},t). \quad (3)$$

Here, $\Psi(\mathbf{r},t)$ is the coherent complex-valued polariton field on the lower-polariton branch in effective mass approximation, with $m_{\mathrm{eff}} = 10^{-4}m_{\mathrm{e}}$ and $m_{\mathrm{e}}$ the free electron mass. Loss of polaritons due to their finite lifetime is included through $\gamma_{\mathrm{c}} = 0.1$ ps$^{-1}$. The repulsive polariton-polariton interaction and resulting cubic nonlinearity in the equation is included with $g_{\mathrm{c}} = 6 \times 10^{-3}$ meV μm$^2$. The external disorder potential is given by $V_{\mathrm{ext}}(\mathbf{r})$. In order to include the relaxation and scattering dynamics for off-resonant optical excitation (Fig. 1b), we assume coupling to two reservoirs, one active reservoir ($n_{\mathrm{A}}$) and one inactive reservoir ($n_{\mathrm{I}}$)[47,50]. The condensate is replenished directly from the active reservoir through a stimulated scattering process with $R = 0.02$ ps$^{-1}$ μm$^2$. The repulsive Coulomb interaction between condensate and excitations in the active reservoir is represented by $g_{\mathrm{r}} = 2g_{\mathrm{c}}$. The density of the active reservoir satisfies

$$\frac{\partial n_{\mathrm{A}}(\mathbf{r},t)}{\partial t} = \tau n_{\mathrm{I}}(\mathbf{r},t) - \gamma_{\mathrm{A}}n_{\mathrm{A}}(\mathbf{r},t) - R|\Psi(\mathbf{r},t)|^2 n_{\mathrm{A}}(\mathbf{r},t). \quad (4)$$

Here, $\gamma_{\mathrm{A}} = 0.15$ ps$^{-1}$ is the polariton loss from the active reservoir. The active reservoir is replenished from the inactive reservoir with $\tau = 0.1$ ps$^{-1}$. The inactive reservoir contains hot excitons excited directly by the external off-resonant pump and obeys the following equation of motion:

$$\frac{\partial n_{\mathrm{I}}(\mathbf{r},t)}{\partial t} = -\tau n_{\mathrm{I}}(\mathbf{r},t) - \gamma_{\mathrm{I}}n_{\mathrm{I}}(\mathbf{r},t) + P(\mathbf{r},t). \quad (5)$$

Here, $\gamma_{\mathrm{I}} = 0.01$ ps$^{-1}$ is the loss from the inactive reservoir and $P(\mathbf{r},t)$ represents the off-resonant pump. As in the experiment, the pump has a frequency far above the excitonic resonance. Here, we assume that the phase information of the optical pulse is lost through the complicated relaxation processes occurring in the real system. In this work, two different optical beams are used for excitation: one is a ring-shaped CW pump for exciting and sustaining a vortex and the other one is a Gaussian-shaped pulse for controlling the topological charge of the vortex (Fig. 1a).

We solve Eqs. (3)–(5) numerically using a split-step Fourier method. The nonlinear contributions are evaluated using the fourth order Runge–Kutta method on a discrete spatial grid. In the simulations, a noisy initial condition with very small amplitude is used for the condensate, $\Psi$, while zero initial conditions are used for the reservoir densities, $n_{\mathrm{A}}$ and $n_{\mathrm{I}}$. The numerical noise used is uniform white noise, generated by random numbers at each grid point in space in both the amplitude and the phase. The noisy initial condition is different for each run of the numerical simulation.

**Microcavity sample.** The sample we investigate is a molecular beam technology-grown planar microcavity based on GaAs with a quality factor of about 20,000 and a Rabi splitting of 9.5 meV. It consists of two distributed-Bragg reflectors (DBR) made of 32 and 36 alternating layers of Al$_{0.2}$Ga$_{0.8}$As and AlAs enclosing a $\lambda/2$ cavity. In the central antinode of the electric field four GaAs quantum wells are placed. The sample is mounted on the cold finger of a helium-flow cryostat to cool it down to the measurement temperature of 17 K. The exciton-cavity detuning is $-4$ meV for all experiments shown in this work.

**Spectroscopy.** For off-resonant optical excitation a CW laser with a wavelength of 735.5 nm (1686 meV) is used. The CW laser is shaped by using a spatial light modulator (SLM) to generate a nominally annular optical potential. For the switching pulses a pulsed titanium–sapphire laser (repetition rate 75.39 MHz) emitting pulses with a duration of approximately 120 fs with the same central wavelength as the CW laser is used. Both beams are focused onto the sample using a microscope objective (numerical aperture 0.4). For real-space imaging a liquid nitrogen-cooled CCD camera placed behind a monochromator (operated in zeroth order) is used. For time-resolved imaging of the OAM, the signal beam is guided through the OAM sorting process and imaged onto a streak camera.

**OAM sorting.** The OAM sorting process works as follows: First, the beam carrying OAM is imaged onto a SLM displaying the transformation phase pattern of the OAM sorter, which transforms the helical phase gradients of OAM states to linear phase gradients. An ideal transformation of the gradients cannot be achieved by a single phase pattern. Accordingly, usually another lens is placed behind the SLM, which performs an optical Fourier transformation of the beam. In the Fourier plane an additional phase correction pattern is placed, which corrects remaining deviations of the wave front from the linear phase gradient. Finally the output light beam is imaged onto a detection device such as a CCD camera using another lens. This lens then focuses different OAM-sorted states with different phase gradients to corresponding spots at different lateral positions on the CCD, where each OAM state is mapped to one detector position. In our experimental implementation[46] the two phase patterns of the OAM sorting process are displayed on two-halves of a single SLM and a concave mirror is used to guide the beam and perform the optical Fourier transformation like a lens would do. The output light beam is then imaged onto the entrance slit of a streak camera, which is opened by 100 μm and aligned along the axis of OAM deflection.

## Data availability

The data that support the findings of this study are available from the corresponding author upon reasonable request.

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

## Acknowledgements

This work was supported by the Deutsche Forschungsgemeinschaft (DFG) through the collaborative research center TRR142 (grant No. 231447078, project A04) and Heisenberg program (grant No. 270619725) and by the Paderborn Center for Parallel Computing, PC². X.M. further ackowledges support from the NSFC (No. 11804064). The Würzburg group acknowledges support by the state of Bavaria.

## Author contributions

X.M. and S.S. conceived the idea and performed the theoretical analysis and numerical simulations. B.B. and M.A. performed the experiments and data analysis. C.S. and S.H. performed the sample growth. X.M., B.B., M.A., R.D., T.M., C.S., S.H. and S.S. discussed the results and contributed to the writing of paper.

## Competing interests

The authors declare no competing interests.
