## [Peer Review File · Nature Communications]

Reviewers' comments:

Reviewer #1 (Remarks to the Author):

The Authors have elaborated a very original work concerning approach to control and switch a localized polariton vortex between opposite vorticity states. Results of numerical simulations are supported by experimental observations of the transitions between states with topological charges +1 and -1.

The work is original and presents novel results, which are of interest not only for polariton physics. I think that this work merits its publication in Nature Communications, but I'd like to suggest that the Authors address the following comments:

1. The work is well written and most steps are clearly explained in a comprehensive and reader accessible language. However, I would suggest adding more details about numerical methods, especially about initial conditions for n_I , n_A used in simulations. I would suggest discussing in more details realization of the noise in the numerical scheme.
2. The question to authors of the paper: Are the authors aware of the work by B. A. Kochetov et.al in PRE 98, 062214 (2018) and arXiv: 1907.06180, where vortex transformation induced by control over external potential are considered and transitions between vortex states are observed. It would be better to add these works in references, in their paper.
3. The authors claim that the same method can be used for multicharged vortices. However, neither experimental nor theoretical demonstrations of this scheme for transformation between higher-order vortices are presented in the paper.
4. It seems that the second term in Eq. (1) should be corrected: the Gaussian control beam with peak density at r_0 is, definitely, not a ring-shaped pulse, as follows from Eq. (1) in its present form.

Reviewer #2 (Remarks to the Author):

The manuscript of X. Ma et al. entitled Realization of all-optical vortex switching in exciton-polariton condensates reports an experimental work related to the observation of vortices in exciton-polariton condensates. The most important aspect of their work is the demonstration of the ability to optically reverse the rotation of their condensates. I believe this work is interesting, e.g. for the field of structured light where optically controlling the spatial phase profile of optical beams is still challenging. However, I would like the authors to address the following points before providing a final recommendation:

1- The approach elaborated by the authors does not allow generating vortices with a unique, well-defined topological charge. When they generate a vortex, they observe contributions from both topological charges (as seen in Fig. 3), with a ratio that depends on local disorder/perturbations. This is a crucial point, and the authors should provide a more quantitative evaluation of how this ratio varies from case to case. Is the case presented in Fig. 3 the best ratio between clockwise and counter-clockwise contributions that they observed or an average one? What is the range of ratios they have measured throughout their sample? How much does this ratio fluctuate in time, e.g. as seen in Fig. 3? The authors should also provide a more precise evaluation of the ratios measured in Fig. 3, as it is very difficult to evaluate solely with a color scale.

2- It is not clear whether the vortices generated are mixed states (each pulse generates a random topological charge, with a preference for one of the two depending on the local disorder) or if they are identical pure states presenting a coherent superposition of two different topological charges (or in-between, i.e. a mix state of two different superpositions). Can the authors distinguish between these two cases? In their simulation of disorder, they calculate a mode which is indeed a superposition of two different topological charges (which is expected, following the breaking of rotational symmetry by disorder). Yet, experimental data do not allow distinguishing between the two cases. Perhaps providing emission spectra would be interesting, as there could be a slight energy change when the vortex is reversed (due to the lift of degeneracy associated to the rotational symmetry breaking). In any cases, this point should be discussed more thoroughly.

3- Is the switching dynamics affected by the ratio between the two topological charges?

4- Overall, I find that the manuscript is lacking data in order to fully appreciate or even understand the underlying physics of the work. This is particularly true for Fig. 3 which is the most important one. As mentioned above, the color maps do not provide a sufficient support to appreciate the ratio between the different topological charges. Line plots (at specific times) should also be provided to better see, quantitatively, the contribution of every topological charge. As mentioned above, providing emission spectra would also be very convenient. Finally, it would be necessary to see more images of the beams

in different conditions: this is crucial to really understand the quality of the angular momentum generated.

Response to Reviewers:

We would like to thank both reviewers for their helpful comments and positive feedback on our work. Our detailed response to each comment is given below. Changes made to the manuscript are highlighted in red color in the main manuscript. In addition to the changes marked, we have added 1D plots in Fig. 3 and deleted Figs. 5(g,h) as with the supplemental material included, the latter did not add much to the discussion anymore. Changes were also made to the captions of Figs. 3 and 5 and a document with Supplemental Material was added.

1. Comments of Reviewer #1:

The Authors have elaborated a very original work concerning approach to control and switch a localized polariton vortex between opposite vorticity states. Results of numerical simulations are supported by experimental observations of the transitions between states with topological charges +1 and -1.

The work is original and presents novel results, which are of interest not only for polariton physics. I think that this work merits its publication in Nature Communications, but I'd like to suggest that the Authors address the following comments:

Many thanks for your kind recommendation and positive comments on the broad impact of our work. We really appreciate it. The answers to your comments are addressed and listed below.

1. The work is well written and most steps are clearly explained in a comprehensive and reader accessible language. However, I would suggest adding more details about numerical methods, especially about initial conditions for n_I , n_A used in simulations. I would suggest discussing in more details realization of the noise in the numerical scheme.

Many thanks for the comment on the numerical method and the related initial conditions. Indeed, the previous version of the manuscript only included the essential information where some of the numerical details were not included. In the revised version, we have added one paragraph in the Methods section to discuss the numerical implementation and the related initial conditions. We would like to note that the results and physical aspects discussed in the manuscript do not depend on the specific choice or parameters of the initial noise in the condensate. Also, the numerical evaluation of the equations of motion is very robust. We hope that you will now find the information given sufficient for reproducibility of our results.

2. The question to authors of the paper: Are the authors aware of the work by B. A. Kochetov et.al in PRE 98, 062214 (2018) and arXiv: 1907.06180, where vortex

transformation induced by control over external potential are considered and transitions between vortex states are observed. It would be better to add these works in references, in their paper.

Many thanks for reminding us of these papers related to our work. We have now cited them in the revised manuscript (see Refs. [36] and [37]). Meanwhile, we also found another related work published recently by Zambon et al. Nature Photonics 13, 283 (2019), which is now also cited as Ref. [38].

3. The authors claim that the same method can be used for multicharged vortices. However, neither experimental nor theoretical demonstrations of this scheme for transformation between higher-order vortices are presented in the paper.

This comment is significant to our work, because we have stated that the switching method in principle also works for higher topological charges in the Conclusions section of our manuscript but we had decided not to show this explicitly. Now, we have numerically calculated the switching dynamics of a topological charge between $m=2$ and $m=-2$ by using the same method, and the results are included and discussed in the Supplemental Material [see Fig. S4]. We believe that this is a very valuable addition to the paper.

4. It seems that the second term in Eq. (1) should be corrected: the Gaussian control beam with peak density at r_0 is, definitely, not a ring-shaped pulse, as follows from Eq. (1) in its present form.

We would like to state that our control pulse has a simple fundamental Gaussian shape instead of a ring shape. This aspect is illustrated in Fig.1 of the manuscript and described in the text. This makes our control scheme even easier to implement in experiment as no complex pulse shaping is needed for the control pulse.

Thank you very much again. We believe that your important comments helped a lot to improve the presentation of our work.

2. Comments of Reviewer #2:

The manuscript of X. Ma et al. entitled Realization of all-optical vortex switching in exciton-polariton condensates reports an experimental work related to the observation of vortices in exciton-polariton condensates. The most important aspect of their work is the demonstration of the ability to optically reverse the rotation of their condensates. I believe this work is interesting, e.g. for the field of structured light where optically controlling the spatial phase profile of optical beams is still challenging. However, I would like the authors to address the following points before providing a final recommendation:

We appreciate very much your encouraging feedback and considering our work for publication in Nature Communications. As detailed below your comments have greatly helped us to further improve the presentation of our work. In addition to the changes made to the main manuscript, we have now also prepared a Supplemental Material document giving more detailed information and addressing your questions.

1- The approach elaborated by the authors does not allow generating vortices with a unique, well-defined topological charge. When they generate a vortex, they observe contributions from both topological charges (as seen in Fig. 3), with a ratio that depends on local disorder/perturbations. This is a crucial point, and the authors should provide a more quantitative evaluation of how this ratio varies from case to case. Is the case presented in Fig. 3 the best ratio between clockwise and counter-clockwise contributions that they observed or an average one?

First, we would like to emphasize that the ratio of the two states is not the main figure of merit of interest to us. We want to demonstrate the switching process, so we want the ratios to be reversed before and after the switching process. In principle the switching still happens even if the two components have a 50/50 distribution. But the switching dynamics would be hard to be distinguished. Secondly, as you correctly noted, this is merely a consequence of the broken rotational symmetry in our system. To give a better feeling of the sensitivity of our main results, we now included data measured at different positions on the sample in the supplemental material. Therefore, we would like to say that the vortex we presented in our work represents a typical (rather than carefully optimized) ratio seen in our measurements. The results shown in the main manuscript have not been carefully optimized and are by no means only observable for one specific position on the sample. In fact, the general switching scheme (albeit not working well everywhere on the sample as illustrated by the new measurements included in the Supplemental Material) is quite robust and can be observed in a similar manner for various different positions on the sample.

What is the range of ratios they have measured throughout their sample?

The ratio for the result shown in Fig. 3 is approximately 2.9 and this is now discussed in more detail in the Supplemental Material. A near perfect vortex ring is also shown in Fig. S2 in the Supplemental Material. The ratio of the two coexisting states ($m=1$ and -1) in experiments depends both on the local sample disorder landscape, pump shape, and contrast achievable in the OAM sorting. We added a comment clarifying this point to the manuscript and provide more information on the role of sample disorder and OAM sorting in the new Supplemental Material. The additional results shown in the supplement also emphasize that a detailed OAM analysis is indeed necessary to characterize the condensate state observed and with the interplay of different effects also the switching does not automatically work better the more

circular the emission of the vortex state looks. For example, the vortex state shown in Fig. S2(b) in the Supplemental Material in the present manuscript is a result of the interplay of local disorder and the imperfect optically induced potential. In this case, we still can observe the switching dynamics, but the switched state is less stable and does not survive on a longer timescale but rather quickly reverts back to the initial state, which is the one preferred in the not strictly rotationally invariant system.

How much does this ratio fluctuate in time, e.g. as seen in Fig. 3?

The ratio of the two states does not significantly fluctuate in time if the system is persistently driven by the CW pump beam and no perturbation, e.g., in form of the control pulse, is applied as in Fig. 3(a,c). From Figs. 3(d) and 3(f) one can also clearly see that on a longer timescale the ratio of the two states is almost the same before and after the application of the control pulse. In the case when “switching” successfully occurs as shown in Fig. 3(b), one can see from Fig. 3(e) that the ratio of the two states is almost reversed.

The authors should also provide a more precise evaluation of the ratios measured in Fig. 3, as it is very difficult to evaluate solely with a color scale.

We agree that this comment is extremely important for the comparison of the results. In the present manuscript, on the one hand, for better visibility we have added the 1D line plots, as shown in Figs. 3(d)-3(f). On the other hand, we have also estimated the ratio (~ 2.9) of the two states based on the OAM sorting data [see the Fig. S3 in the Supplemental Material and related discussion], which shows reasonable agreement with our numerical results. However, we would like to note that for this result neither the optical setup nor the sample in the experiments were optimized to achieve optimum switching performance. We would like to emphasize again here that the value of the ratio is not really our figure of merit as we mentioned above. Most important is that the reversion of the ratio clearly demonstrates the switching dynamics.

2- It is not clear whether the vortices generated are mixed states (each pulse generates a random topological charge, with a preference for one of the two depending on the local disorder) or if they are identical pure states presenting a coherent superposition of two different topological charges (or in-between, i.e. a mix state of two different superpositions). Can the authors distinguish between these two cases? In their simulation of disorder, they calculate a mode which is indeed a superposition of two different topological charges (which is expected, following the breaking of rotational symmetry by disorder). Yet, experimental data do not allow distinguishing between the two cases. Perhaps providing emission spectra would be interesting, as there could be a slight energy change when the vortex is reversed (due to the lift of degeneracy associated to the rotational symmetry breaking). In any cases, this point should be discussed more thoroughly.

This comment is extremely important for improving our presentation. In order to clarify this aspect we have performed correlation measurements [see Fig. S1 in the Supplemental Material] instead of the measurement of the emission spectra as suggested by the referee. The results obtained indeed provide solid evidence that the interpretation as used in the manuscript is correct and the two modes ($m=1$ and $m=-1$) appear in the OAM-sorting results due to their coexistence instead of the creation of a single OAM mode in each individual excitation cycle. We would like to thank the referee for pushing us to include this important piece of information in the manuscript.

Although the OAM sorting is the main experimental technique for the observation of the switching dynamics in our work, we now also mention in the manuscript, that measurement of emission spectra could also be helpful to observe signatures of vortex switching between different states. In this context we have also added the following important references to the manuscript: Zambon et al, Nature Photonics 13, 283 (2019) and OL 44, 4531 (2019) as Refs. [38,46], respectively. We chose not to show spectrally resolved data, as in general, the high temporal resolution achieved and needed in our optical control measurements and a sufficient spectral resolution would exclude each other.

3- Is the switching dynamics affected by the ratio between the two topological charges?

We do not see any significant difference in the dynamics as long as the approximation of a slightly perturbed circular potential is realized (also compare with the results shown in Fig.S2). If one distorts the potential more significantly, also the dynamics will change as the potential changes and the picture of an approximately circular vortex flow becomes invalid. This is exactly why we choose to work at only slightly perturbed potentials.

4- Overall, I find that the manuscript is lacking data in order to fully appreciate or even understand the underlying physics of the work. This is particularly true for Fig. 3 which is the most important one. As mentioned above, the color maps do not provide a sufficient support to appreciate the ratio between the different topological charges. Line plots (at specific times) should also be provided to better see, quantitatively, the contribution of every topological charge. As mentioned above, providing emission spectra would also be very convenient. Finally, it would be necessary to see more images of the beams in different conditions: this is crucial to really understand the quality of the angular momentum generated.

Many thanks for your important suggestions to further improve the clarity and presentation of our work. We have significantly revised our manuscript and attached a supplemental material for detailed discussion. As you suggested, we have added

the 1D line plots to Fig.3, now show different vortex profiles measured at different sample positions, and explicitly demonstrate the coexistence of the $m=+1$ and $m=-1$ state. We hope that you will appreciate our effort and will find that the additional data give the extra support needed to recommend publication of our paper.

The new results and materials provided in the resubmitted files are listed below:

In the main manuscript:

In response to the Reviewers' comments, various smaller changes were made to the text in the main manuscript (highlighted in red). In addition as suggested by Reviewer #2, we have added 1D line plots in Fig. 3. Also the figure caption was modified accordingly. We have removed panels (g) and (h) from Figure 5 since with the additional numerical analysis of vortex states including disorder now included in the manuscript, these plots did no longer add any valuable information.

In the Supplemental Material we included the following information to further support the results and discussion in the main text:

1. We have measured the correlation of the two states ($m=1$ and $m=-1$), evidencing that the states shown in Fig. 3 are the coexistence of them for each excitation. [see Fig. S1].
2. We have added two new measurements at different positions of the sample to demonstrate the influence of the disorder potential on the switching dynamics. [see Fig. S2].
3. We have estimated the related ratio of the coexisting two states ($m=1$ and $m=-1$) from the OAM sorting data in Fig. 3. [see Fig. S3] and compare with the relevant numerical analysis.
4. Switching of higher-order ($|m|=2$) vortices within the same scheme is realized numerically and shown.

We believe that the presentation in the resubmitted manuscript is now significantly improved according to the Reviewers' comments. We hope that the Reviewers' will appreciate our efforts including the extra measurements performed and will now recommend our manuscript for publication in Nature Communications.

Reviewers' comments:

Reviewer #1 (Remarks to the Author):

The revised manuscript and response on the Referees' comments addressed mostly my comments and concerns. There are, however, several minor points, which must be accounted.

4. It seems that the second term in Eq. (1) should be corrected: the Gaussian control beam with peak density at r_0 is, definitely, not a ring-shaped pulse, as follows from Eq. (1) in its present form.

We would like to state that our control pulse has a simple fundamental Gaussian shape instead of a ring shape. This aspect is illustrated in Fig.1 of the manuscript and described in the text. This makes our control scheme even easier to implement in experiment as no complex pulse shaping is needed for the control pulse.

1. Obviously, the control pulse is NOT a ring-shaped pulse, as it clearly seen both from figures and from the context in the manuscript. However, Eq. (1) still contains the second term, which is, definitely, an axially-symmetric (ring-shaped) function. This seems to be typos.

2. I suggest rewriting Eq. (2), which describes the topological charge m . It easy to verify that for non-stationary states present definition of m corresponds to a complex-valued function. I would suggest using a common definition of the angular momentum per particle, which gives a real-valued topological charge.

3. The question to authors of the paper: Are the authors aware of the work by G. Nardin et.al. Journal of Nanophotonics 5, 053517 (2011) (DOI: 10.1117/1.3609825). It is relevant to add a brief discussion of this experimental observation of oscillation between orbital momentum states in a semiconductor microcavity. Please also update the Ref. [37] by the published version (DOI:10.1103/PhysRevE.100.062202).

In conclusion, I recommend publishing this work in Nature Communications after minor revision.

Reviewer #2 (Remarks to the Author):

The authors have well addressed my comments. I now recommend the publication of their manuscript.

Response to Reviewers:

We appreciate very much your recommendation as well as your important comments for further improving our presentation. All your comments are now addressed and our detailed response is listed below.

Comments of Reviewer #1:

1. Obviously, the control pulse is NOT a ring-shaped pulse, as it clearly seen both from figures and from the context in the manuscript. However, Eq. (1) still contains the second term, which is, definitely, an axially-symmetric (ring-shaped) function. This seems to be typos.

Thank you very much for pointing out the typo in Eq (1) again. This was an unfortunate mistake and we regret not having seen it before but it has been corrected now.

2. I suggest rewriting Eq. (2), which describes the topological charge m . It easy to verify that for non-stationary states present definition of m corresponds to a complex-valued function. I would suggest using a common definition of the angular momentum per particle, which gives a real-valued topological charge.

We fully agree with this comment. We have used this definition of topological charge (angular momentum per particle) in one of our recent works [Xue et al., arXiv:1907.00383]. We also checked that these two methods agree in our case. However, as you kindly mentioned, the definition of the angular momentum per particle is more common and suitable for general readers. In the revised manuscript, we have updated it.

3. The question to authors of the paper: Are the authors aware of the work by G. Nardin et.al. Journal of Nanophotonics 5, 053517 (2011) (DOI: 10.1117/1.3609825). It is relevant to add a brief discussion of this experimental observation of oscillation between orbital momentum states in a semiconductor microcavity. Please also update the Ref. [37] by the published version (DOI:10.1103/PhysRevE.100.062202).

The related work has been cited and discussed briefly in the current manuscript and the Ref. [37] has been updated. Thank you very much for reminding us.

Comments of Reviewer #2:

The authors have well addressed my comments. I now recommend the publication of their manuscript

We appreciate very much your recommendation.

REVIEWERS' COMMENTS:

Reviewer #1 (Remarks to the Author):

The authors have addressed my comments mostly. I can recommend the publication of their manuscript after minor correction.

The definition of the topological charge m used in Eq. (2) still gives the complex-valued function. It seems to be a typo. Indeed, the angular momentum operator includes imaginary unit:

$$\vec{L} = -i\hbar \vec{r} \times \nabla.$$

Please verify the definition of the angular momentum L_z (see e.g PRB 91, 184518 (2015)).

Response to Reviewers:

We appreciate very much your recommendation as well as your important comment for pointing out the remaining typo in our manuscript.

Comments of Reviewer #1:

1. The authors have addressed my comments mostly. I can recommend the publication of their manuscript after minor correction.

The definition of the topological charge m used in Eq. (2) still gives the complex-valued function. It seems to be a typo. Indeed, the angular momentum operator includes imaginary unit:

$$\vec{L} = -i\hbar \vec{r} \times \nabla.$$

Please verify the definition of the angular momentum L_z (see e.g PRB 91, 184518 (2015)).

Indeed, the imaginary symbol ‘-i’ was missing in the previous version of our manuscript for the definition of the angular momentum L_z , so that in the latest version of our manuscript, we have corrected it.